# HoneyLite: A Lightweight Honeypot Security Solution for SMEs

**DOI:** 10.3390/s25165207

**Published:** 2025-08-21

**Authors:** Nurayn AlQahtan, Aseel AlOlayan, AbdulAziz AlAjaji, Abdulaziz Almaslukh

**Affiliations:** Information System Department, King Saud University, Riyadh 11451, Saudi Arabia; 445205744@student.ksu.edu.sa (N.A.); 445205825@student.ksu.edu.sa (A.A.); aalmaslukh@ksu.edu.sa (A.A.)

**Keywords:** cybersecurity, honeypots, malware, resource efficiency, SMEs

## Abstract

Small and medium-sized enterprises (SMEs) are increasingly targeted by cyber threats but often lack the financial and technical resources to implement advanced security systems. This paper presents HoneyLite, a lightweight and dynamic honeypot-based security solution specifically designed to meet the constraints and cybersecurity needs of SMEs. Unlike traditional honeypots, HoneyLite integrates real-time network traffic analysis with automated malware detection via the VirusTotal API, enabling it to identify a wide range of cyber threats, including TCP scans, FTP/SSH intrusions, ICMP flood attacks, and malicious file uploads. Developed using open-source tools, the system operates with minimal resource overhead and is validated within a simulated virtual environment. It also generates detailed threat reports to support incident analysis and response. By combining affordability, adaptability, and comprehensive threat visibility, HoneyLite offers a practical and scalable solution to help SMEs detect, analyze, and respond to modern cyberattacks in real time.

## 1. Introduction

SMEs are increasingly targeted by cyber threats due to their growing digital presence and limited cybersecurity preparedness [1,2]. Despite their critical role in economic development, many SMEs lack the financial, technical, and human resources to implement and maintain advanced security infrastructures [3]. Factors such as limited budgets, lack of awareness, overconfidence, and a short-term focus further exacerbate their vulnerability to cyberattacks [4]. Consequently, the financial and operational impacts of cyber incidents—ranging from data breaches and service disruptions to reputational damage and legal liabilities—are often more severe for SMEs than for larger organizations [5].

Recent studies have shown that SMEs are particularly vulnerable to attacks such as Denial-of-Service (DoS), unauthorized access, and malware infections [6]. These threats are not only highly disruptive but are also increasingly automated, targeted, and cost-effective for attackers to execute [7]. Moreover, SMEs typically lack formal containment and recovery strategies, making the need for proactive and intelligent threat detection mechanisms all the more urgent [3].

According to Statista [8] data, cyber incidents have consistently ranked among the top risks for small enterprises worldwide from 2018 to 2023. As illustrated in Table 1, the data reflects a sustained and visually evident trend in which cybersecurity threats remain a leading and growing concern for SMEs—underscoring the urgent need for affordable and intelligent defense solutions to enhance resilience in an increasingly hostile digital environment.

One promising approach is the use of honeypots—deceptive [9] systems designed to lure attackers, monitor their behavior, and gather forensic data without risking actual business assets [10]. Traditional honeypot solutions, however, often require significant resources and tend to focus narrowly on either network intrusion or malware detection, limiting their usefulness for SMEs with constrained capabilities [11].

To address these challenges, we propose HoneyLite: a lightweight, dynamic, and resource-efficient honeypot security system specifically designed for SMEs. HoneyLite combines real-time network traffic monitoring with automated malware detection using the VirusTotal API [12]. It is capable of identifying a wide range of cyber threats, including TCP port scans, FTP/SSH brute-force attempts, ICMP flood attacks, and malicious file uploads. Developed with open-source tools and tested in a controlled virtual environment, HoneyLite operates with minimal overhead and produces actionable threat reports that aid in incident response. The key contributions of this paper are as follows:HoneyLite System Design: We design and implement HoneyLite, a novel honeypot-based security framework tailored for the resource constraints and threat profile of SMEs.Dual-Layer Threat Detection: Unlike traditional systems, HoneyLite integrates both real-time network intrusion detection and automated malware scanning using the VirusTotal API.Resource Efficiency: The system is built using lightweight, open-source tools that ensure low resource consumption and ease of deployment on modest hardware.Realistic Validation: HoneyLite is evaluated in a controlled virtual environment simulating real-world attacks, demonstrating its effectiveness in detecting multiple threat vectors.Actionable Reporting: The system generates comprehensive reports summarizing detected threats, including source IPs, service ports, attack types, and malware signatures.

These contributions establish HoneyLite as a practical and scalable cybersecurity solution for SMEs seeking to enhance their cyber resilience without significant investment. Figure 1 illustrates the virtual environment for HoneyLite deployment, which simulates attacker, victim, and honeypot nodes.

The rest of this paper is structured as follows: Section 2 presents a review of related work, focusing on existing honeypot systems and their applicability to SMEs. Section 3 outlines the aim and specific objectives of this research. Section 4 details the proposed methodology, including system architecture, components, and implementation phases. Section 5 discusses the experimental setup, testing procedures, and key results obtained from the simulated environment. Section 6 explores potential enhancements and future work. Finally, Section 7 concludes the paper by summarizing the main findings and contributions of the HoneyLite system.

## 2. Literature Review

This section describes the current landscape of malware and network anomaly detection strategies relevant to SMEs. It provides a critical evaluation of both traditional and emerging approaches, with emphasis on dynamic honeypot technologies.

### 2.1. Traditional Methods for Malware and Network Anomaly Detection in SMEs

Prior to the widespread adoption of honeypot-based solutions, SMEs commonly relied on conventional cybersecurity tools for network and malware detection. These methods included signature-based intrusion detection systems (IDS), firewalls, antivirus solutions, and heuristic algorithms [1,13,14]. While widely deployed due to their ease of installation and relatively low cost, these approaches are often reactive in nature and struggle to detect previously unseen attack vectors, such as zero-day exploits or sophisticated, automated attacks that frequently target SMEs [15]. Furthermore, such systems can generate high rates of false positives and may be difficult for resource-constrained SMEs to manage effectively without dedicated IT staff or external support [16].

Traditional systems are also often hindered by a lack of contextual awareness and behavioral analysis, limiting their ability to detect advanced persistent threats or polymorphic malware [17]. SMEs tend to underestimate their exposure to cyber risk, frequently resulting in suboptimal allocation of resources to cyber defense. A systematic review [18] highlights that small businesses often lack formal risk management and typically use generalized security controls without adaptation for their specific operational environments. Moreover, recent studies show that existing antivirus and IDS tools are susceptible to evasion techniques and may not provide timely alerts for attack campaigns leveraging novel malware samples [19]. To supplement traditional tools, some SMEs have adopted threat intelligence feeds and external security assessment services, but these solutions may introduce new costs and require skilled staff to interpret findings [17,20]. The persistence of skills shortages and limited cybersecurity training in SMEs continues to impede the effectiveness of these conventional defenses [21].

### 2.2. Recent Advances in Dynamic Honeypot Approaches

There has been growing interest in dynamic honeypot technologies in recent years, particularly within the SME sector, as a means to address increasingly sophisticated and evolving cyber threats [22,23,24,25]. Dynamic honeypots offer a proactive defense mechanism by luring attackers, recording their activities, and gathering forensic data.

A notable work in this area is the HoneyTrack system, which creates database decoys designed to attract and engage attackers, thus enabling detailed analysis of their behavior [22]. HoneyTrack uses Docker and Python for a flexible deployment, yet its implementation is limited by a lack of real-world validation and a narrow focus on specific asset types. In the realm of Internet of Things (IoT) security, reinforcement learning has been combined with honeypot technology to detect sophisticated attacks such as Man-in-the-Middle (MitM) and Distributed Denial of Service (DDoS), demonstrating the potential of integrating artificial intelligence (AI) with deception techniques [23].

Low-interaction honeypots have also been used to enhance the security posture of medium-to-large organizations by enabling deeper insight into attacker methodologies and informing more effective security protocols [24]. Honeypots have further been leveraged to simulate and analyze DDoS scenarios, thereby supporting the evaluation and development of new mitigation strategies [25]. The targeted study of specific protocols, such as SSH and Telnet, has benefited from using honeypots to reveal attacker tools and exploitation techniques [26].

To address scalability and resilience, recent papers have integrated honeypots with cloud services like AWS Lambda and Elastic Load Balancing, achieving more dynamic and distributed security solutions—though this approach introduces new challenges regarding configuration and operational complexity [27]. Honeypots have also been employed to enhance the security of Industrial Control Systems (ICS), highlighting both their efficacy in real-world scenarios and the difficulties arising due to legacy architectures [28]. Additional studies have demonstrated how honeypot deployments can capture a range of DDoS attack methods, such as Slowloris and GoldenEye, albeit with observed limitations in detecting all forms of such attacks [29].

Recent literature also reports on novel lightweight honeypot frameworks specifically designed for the constraints of SMEs. For example, this work [30] has developed a resource-efficient distributed honeypot system capable of detecting port scanning, IoT-specific attacks, and ransomware propagation, achieving a low system footprint while maintaining detection accuracy [31]. Other works have explored hybrid honeypot architectures that mix low- and high-interaction honeypots for improved coverage with manageable cost [32]. The deployment of virtualized honeypots on containers or microservices platforms, as demonstrated in [33], has further reduced deployment complexity for SMEs.

Despite rapid innovation in honeypot technologies, several persistent obstacles hinder their widespread adoption, particularly within SMEs. These include issues related to scalability, real-time threat correlation, ease of deployment for users without specialized expertise, ongoing maintenance burdens, and vulnerabilities associated with honeypot exposure and evasion [34]. To address these limitations, contemporary research increasingly emphasizes the need for automation, simplified configuration interfaces, and seamless integration with open-source monitoring and alerting tools as essential components of SME-appropriate honeypot solutions [35]. While dynamic honeypots have demonstrated strong potential for proactive threat detection, their adoption in SME environments remains constrained due to factors such as lack of real-world validation, narrow detection scopes, and excessive resource consumption. In response to these challenges, this paper proposes HoneyLite —a lightweight, open-source honeypot framework specifically engineered for SMEs. By combining dynamic network monitoring, automated malware detection via the VirusTotal API, and structured reporting with minimal deployment complexity and system overhead, HoneyLite offers a cost-effective and practical cybersecurity solution tailored to the resource and operational constraints of small and medium-sized enterprises.

### 2.3. Emerging Threats: Ransomware and Supply Chain Attacks

Recent developments have seen a marked increase in ransomware variants and supply chain attacks, underscoring the urgency for updated detection strategies. For instance, 2024 witnessed a 123% rise in ransomware activity with dozens of new attacker groups emerging [36]. Simultaneously, supply chain attacks became nearly daily occurrences, exposing broad vulnerabilities within software ecosystems [37]. Honeypot research has begun to address these evolving threats; recent work proposes file-based honeypots integrated with eBPF-based monitoring for early ransomware detection under zero-trust architectures [38]. Similarly, the HoneyWin platform presents a high-interaction Windows honeypot tailored for enterprise IT infrastructure and ransomware-related behavior analysis [33].

## 3. Aim and Objectives

### 3.1. The Aim

The aim of this project is to develop HoneyLite, a lightweight and dynamic honeypot system tailored for SMEs, to enhance their cybersecurity defenses through real-time threat detection and malware analysis with minimal resource requirements.

### 3.2. Objectives

The following is the list of objectives to accomplish:Develop a system capable of detecting a wide range of cyber threats, including TCP scans, FTP/SSH intrusions, ICMP floods, and unauthorized access attempts.Integrate automated malware detection using the VirusTotal API to identify malicious file uploads and injection attempts.Design a resource-efficient solution that can be easily deployed within SME environments using lightweight, open-source tools.Generate detailed threat reports to support effective incident analysis and response.

## 4. Methodology

This paper adopts a honeypot-based approach to detect cyber threats targeting SMEs. The proposed system, HoneyLite, is designed to be both resource-efficient and dynamic, allowing real-time detection of network intrusions and malware injection attempts with minimal hardware requirements.

### 4.1. Honeypot Overview

A honeypot is composed of two fundamental components: decoys and captors. Its core purpose is to expose fake vulnerabilities by emulating real systems, thereby attracting malicious actors and enabling detailed observation of their behavior. Honeypots open various service ports to simulate legitimate services, thereby enticing unauthorized access attempts for security investigation. By engaging attackers, honeypots consume adversarial resources, delay further attacks, and collect valuable forensic data for analysis and response [39]. We clarify here that HoneyLite differs from existing honeypots by focusing on lightweight deployment and minimal system footprint tailored for SMEs.

The HoneyLite methodology is implemented through four main phases: setup, execution, monitoring, and reporting. Each phase is designed to optimize detection capabilities while maintaining efficiency and simplicity, particularly for deployment in SME environments. For clarity, we explicitly describe how each phase connects to the underlying data structures and analysis tasks (see Figure 2).

### 4.2. Setup Phase

The system continuously monitors network traffic and file system activity using structured data representations to detect anomalies and threats. Two primary dictionaries are used to track the frequency of network packets:$$C_{\mathrm{tcp}}\left(\mathrm{src},\mathrm{dst}\right)\;\mathrm{and}\;C_{\mathrm{icmp}}\left(\mathrm{src}\right)$$
where src and dst represent the source and destination IP addresses, respectively. These counters help identify abnormal traffic patterns that may indicate scanning or flooding attacks.

To log detailed information about potential attack attempts, another dictionary is maintained:$$A_{d}=\left(\mathrm{src},\mathrm{dst},\mathrm{service},\mathrm{flag}\right)\to \left(\mathrm{start}\_\mathrm{time},\mathrm{packet}\_\mathrm{count}\right)$$

This captures essential metadata associated with each suspicious activity, including protocol used and connection timing.

The dictionaries $C_{tcp}$ and $C_{icmp}$ help identify abnormal patterns in network activity. When the recorded values exceed certain thresholds, such as repeated TCP requests to a specific destination or a high number of ICMP messages from the same source, a corresponding entry is added to $A_{d}$. This entry summarizes the suspicious behavior by storing relevant metadata like source, destination, service type, and time information. In this way, $C_{tcp}$ and $C_{icmp}$ act as supporting structures that trigger the creation of more detailed attack records in $A_{d}$, which are later used in the reporting phase. asaWe explicitly note here that $A_{d}$ consolidates these abnormal events into structured logs, ensuring that the reporting phase produces a comprehensive record of both traffic anomalies and file-based threats.

For detecting and analyzing malicious files, the system maintains a list of tuples containing malware scan results:$$M=\{\left(f_{i},h_{i},c_{i}\right)|i=1,2,\ldots ,n\}$$
where $f_{i}$ is the path of the $i^{th}$ file, $h_{i}$ is its MD5 hash, and $c_{i}$ denotes the number of malware signatures identified. This structure supports efficient look-up and correlation of malicious artifacts with incoming attack traffic. In practice, the integration of *M* with $A_{d}$ allows correlating suspicious traffic sessions with malware samples uploaded during the same interaction, thereby improving attribution and forensic accuracy.

### 4.3. Execution Phase

In the execution phase, the honeypot system begins actively listening to network traffic and monitoring file system events. Packet sniffing is performed using the Scapy library, and all incoming packets are processed by a packet_handler function.

Let $P=\{p_{1},p_{2},\ldots ,p_{n}\}$ represent the set of captured packets over time *t*. For each TCP packet $p_{i}$ with source IP src and destination port dst, the following operation is performed:$$C_{\mathrm{tcp}}\left(\mathrm{src},\mathrm{dst}\right)\;+=\;1$$

ICMP traffic is similarly monitored. If the number of ICMP packets from a given source exceeds a predefined threshold $T_{flood}=10$ packets per second, the system flags it as a potential ICMP flood attack:$$\mathrm{If}\;C_{\mathrm{icmp}}\left(\mathrm{src}\right)\geq T_{\mathrm{flood}},\;\mathrm{then}\;\mathrm{flag}\;\mathrm{as}\;\mathrm{DoS}\;\mathrm{attack}$$

Simultaneously, file uploads to the honeypot are detected using FileSystemEventHandler from watchdog. Once a new file is identified, its MD5 hash is computed and submitted to the VirusTotal API through the check_file_virustotal function for malware analysis. By combining traffic-based anomaly detection and file-based analysis in real time, the execution phase ensures that the honeypot captures both network-level and host-level attack vectors.

### 4.4. Monitoring Phase

The system continuously monitors both network activity and file system changes. TCP connection attempts, ICMP traffic, and new file events are captured in real time to maintain an up-to-date view of suspicious behaviors and potential breaches. To improve clarity, we emphasize that this phase acts as the bridge between execution and reporting, ensuring that raw data collected in real time is preserved, filtered, and prepared for structured reporting.

### 4.5. Reporting Phase

In the final phase, the system compiles all collected data into a structured report. Let $A_{d}$ represent the set of attack records and $M_{d}$ the set of malware detection instances. The complete report *R* is defined as$$R=\overset{n}{\underset{i=1}{⋃}}A_{d_{i}}\cup M_{d}$$

Each attack detail is represented as$$A_{d}=\left(\mathrm{service},\mathrm{flag},\mathrm{src},\mathrm{port},\mathrm{start}\_\mathrm{time},\mathrm{end}\_\mathrm{time},\mathrm{packet}\_\mathrm{count}\right)$$

Each malware detection record is defined as$$M_{d}=\left(\mathrm{file}\_\mathrm{path},\mathrm{md}5\_\mathrm{hash},\mathrm{detection}\_\mathrm{count}\right)$$These reports provide SMEs with actionable insights for security assessment, incident response, and future defense planning. We highlight that the reporting phase synthesizes network anomalies ($A_{d}$) and file-level detections ($M_{d}$) into a single, SME-friendly report format, which can be directly used by IT managers for incident response without requiring advanced cybersecurity expertise.

### 4.6. Implementation Details

The implementation of the HoneyLite system is carried out in a virtualized environment configured on a high-performance host machine. Oracle VM VirtualBox version 6.1.42 [40] is used to create and manage multiple virtual machines (VMs), enabling isolated simulation of both attacker and target systems [24]. The honeypot component is deployed on a Windows 10 VM, equipped with the npcap [41] library to support low-level packet capture and analysis. The attacker system is emulated using Kali Linux [42], which provides a rich suite of penetration testing tools to simulate realistic cyber threats [39]. The core honeypot logic is implemented in Python (version 3.12.2, 64-bit, build: python-3.12.2-amd64) leveraging open-source libraries including scapy [43] for packet analysis, watchdog [44] for monitoring file system events, schedule [45] for periodic task execution, and requests [46] for HTTP-based API integration. This software stack enables HoneyLite to operate efficiently while maintaining flexibility and ease of deployment within SME environments. To further improve reproducibility, we clarify that the HoneyLite system can be deployed using a standard personal laptop (e.g., 8 GB RAM, quad-core CPU, and 256 GB storage), making the setup accessible and practical for SMEs without requiring specialized hardware.

## 5. Results and Discussion

This section presents the evaluation results of the HoneyLite system and discusses their relevance in the context of improving cybersecurity for SMEs. The system’s effectiveness was assessed in a controlled, simulated environment, enabling the observation of its behavior under various attack scenarios. For transparency and reproducibility, the complete implementation code of the monitoring engine, including the logging of packet flags, frequency, source and destination IPs, and targeted ports, is available at our public repository: https://github.com/azajaji/HoneyLite (accessed on 17 August 2025).

### 5.1. Testing Environment

The evaluation environment was deployed using Linux-based virtual machines within a VirtualBox hypervisor. This setup ensured reproducibility and proper isolation of each system. Three distinct VMs were configured, each with a dedicated role: Attacker, Victim, and Honeypot.

The Attacker machine was responsible for generating malicious traffic and simulating adversarial behavior. The Victim represented a standard service host targeted by attacks, while the Honeypot acted as the deceptive system designed to capture and analyze attack attempts. All network communication between these components was routed through controlled virtual network interfaces, ensuring secure isolation while enabling accurate traffic redirection and monitoring.

This virtualized architecture was chosen to balance realism with practicality, allowing experiments to be conducted in a safe and reproducible environment. Furthermore, the lightweight setup makes it feasible for deployment in SME contexts, where resources are typically constrained but security monitoring remains essential.

### 5.2. Monitoring and Analysis

During the monitoring phase, multiple attack strategies were simulated to evaluate the effectiveness of the proposed HoneyLite honeypot system. These included initiating FTP and SSH connection attempts, establishing a reverse TCP connection, and launching an ICMP flood to simulate a Denial-of-Service (DoS) attack, as illustrated in Figure 3. This configuration simulates a more interactive threat landscape, exposing the honeypot to a variety of attack vectors and allowing for robust data collection and analysis.

While this environment enables HoneyLite to observe a broad range of malicious behaviors, the classification of the honeypot as low-interaction or high-interaction depends on the degree of engagement with attackers, not merely the diversity of attack types it can detect.

HoneyLite is designed to function on two primary fronts: real-time network monitoring and file-based malware detection. The system continuously observes network traffic for suspicious activity while simultaneously analyzing downloaded files for potential malware. The following subsections detail these core components and their respective functionalities.

To ensure accurate and efficient detection, HoneyLite leverages lightweight, open-source libraries that minimize performance overhead while maximizing threat visibility. The monitoring engine operates continuously, logging relevant metadata from observed traffic such as packet flags, frequency, source and destination IPs, and targeted ports. Concurrently, file-based events are captured via a dedicated handler, enabling the system to detect and analyze any suspicious file uploads or downloads in real time. This dual approach enhances the honeypot’s capability to correlate network anomalies with file-based indicators of compromise, providing SMEs with deeper situational awareness and actionable insights into ongoing threats.

#### 5.2.1. Network Monitoring

HoneyLite employs the scapy library to perform real-time network traffic monitoring and analysis, focusing on TCP and ICMP protocols. By capturing and inspecting packet-level data, the system identifies anomalous communication patterns that may indicate malicious activity. These detection capabilities are crucial for SMEs, where early threat identification can significantly reduce potential damage. Specific attack types and their corresponding detection methods are outlined below:TCP Attacks: These attacks exploit vulnerabilities in the TCP handshake or data transmission processes to disrupt or manipulate network communication [47]. HoneyLite inspects TCP flags to detect scanning behavior, such as SYN scans, which are indicative of reconnaissance attempts. For instance, a TCP packet with the SYN flag targeting an open port may signify an unauthorized connection attempt. Detection results are shown in Figure 4. Repeated SYN attempts from a single source IP are flagged for further investigation.FTP Attacks: File Transfer Protocol (FTP) attacks aim to exploit insecure authentication and data transfer mechanisms to gain unauthorized access or manipulate files [48]. HoneyLite identifies such threats by monitoring TCP traffic directed to port 21, the standard FTP service port. It also records the frequency and timing of such connection attempts to identify brute-force or scripted attack behavior. Figure 4 illustrates observed FTP probing behavior.SSH Attacks: Secure Shell (SSH) services are often targeted for brute-force login or remote command execution [49]. HoneyLite detects SSH-based threats by monitoring connection attempts on port 22, particularly those with SYN flags, which suggest initial handshake attempts. By correlating repeated access patterns from the same source, the system highlights potential compromise attempts. Detection traces are also presented in Figure 4.DoS Attack Detection: Denial-of-Service (DoS) attacks aim to exhaust system resources by overwhelming the target with excessive traffic [49]. HoneyLite identifies potential ICMP flood attacks by tracking the rate of ICMP packets per source. If the volume exceeds a predefined threshold of 10 packets per second, the system flags it as a possible DoS attempt. This detection mechanism is particularly effective for identifying early stages of volumetric attacks. The detection outcome is demonstrated in Figure 5.

#### 5.2.2. File Monitoring

HoneyLite incorporates file-level monitoring and malware detection capabilities by integrating the VirusTotal API, thereby enhancing its threat detection beyond network-level analysis. The VirusTotal API allows programmatic access to malware scanning services, enabling the system to upload files, retrieve scan reports, and analyze them without relying on the web interface [12].

Malware File Detection: HoneyLite employs the watchdog library to continuously observe a specified directory—typically representing a download or drop location for potentially malicious files. This simulates a scenario in which attackers attempt to upload or deliver malware to the honeypot system. Upon detecting the creation of a new file, the system computes its MD5 hash and queries the VirusTotal database via API to determine whether the file is associated with known malware signatures.If no threats are identified, the system outputs the message “No Malware Detected for files”, as illustrated in Figure 6. This approach enables real-time identification of known malicious artifacts and contributes to HoneyLite’s multi-layered threat detection capabilities.Beyond simple hash-based detection, HoneyLite can be extended to support file content inspection, behavior-based tagging, and integration with sandbox environments for dynamic analysis. This allows the system to flag files that may be previously unknown but exhibit suspicious characteristics. Regular polling and efficient event-handling ensure that no file changes are missed, and every file is systematically evaluated for risk, improving the robustness of the monitoring mechanism.

#### 5.2.3. Reporting and Output Generation

Upon completion of the monitoring process, HoneyLite generates a comprehensive report summarizing all detected network and file-based threats, as shown in Figure 7. This report is automatically produced when the monitoring script is terminated by the user, ensuring a streamlined and user-controlled reporting mechanism.

The report encapsulates critical information related to observed malicious activities, including the start time of the attack, source and destination IP addresses, destination ports, and the total number of packets associated with each event. In addition to network-related data, HoneyLite logs the results of its file-based malware detection efforts. This includes details such as the file path, computed MD5 hash, and the number of positive detections retrieved from the VirusTotal database, along with the timestamp of the detection and API response metadata.

By consolidating both network and file system threat data, the reporting module provides actionable insights for security analysis, forensic investigations, and incident response, thereby enhancing the system’s overall effectiveness in threat detection and mitigation within resource-constrained SME environments.

### 5.3. Privacy, Legal, and Compliance Considerations in Honeypot Deployment

While honeypots are valuable for detecting and mitigating cyber threats, they also raise concerns around privacy, legality, and compliance. Capturing attacker data or traffic may involve processing personal information, which under regulations like GDPR requires strict controls on collection, consent, and lawful use [50]. Non-compliance risks legal penalties and reputational damage. Standards such as ISO/IEC 27001 [51] and NIST SP 800-53 [52] stress risk management, data minimization, and transparency. For SMEs, balancing proactive security with regulatory obligations is critical. Recommended practices include anonymizing captured data, limiting retention, and restricting access [53]. Ethical deployment further requires avoiding the capture of legitimate user data [54]. Thus, honeypots for SMEs should be designed with compliance-by-design principles, embedding privacy safeguards and aligning with legal frameworks to ensure both effectiveness and lawful operation.

## 6. Limitations

While the presented evaluation demonstrates the functional validity of HoneyLite in a controlled virtualized environment, we acknowledge several limitations in scope. First, due to the simplified setup, we were unable to provide comprehensive quantitative metrics such as detection rate, false positive/negative rate, and resource consumption. Likewise, direct comparative experiments with more mature honeypot frameworks were not performed, as the lightweight design of HoneyLite does not yet offer a fair basis for such evaluation. Second, although the proof-of-concept tests suggest that HoneyLite can adequately handle simulated attack traffic, its resilience against high-bandwidth or distributed attack scenarios remains untested.

## 7. Future Work

Future enhancements to the HoneyLite honeypot system can significantly strengthen its effectiveness and adaptability in addressing evolving cybersecurity threats. The following avenues are proposed for further development:Expanding Detection Capabilities: Future iterations of HoneyLite could incorporate advanced mechanisms to detect a broader range of cyberattacks beyond the current scope. This includes capabilities to identify emerging threats such as zero-day exploits, brute-force login attempts, and web-based attacks like SQL injection and cross-site scripting (XSS). Continued research into evolving attack vectors will be essential for maintaining the system’s relevance and resilience. To specifically overcome the limitations of relying on static signature databases like VirusTotal, we plan to integrate dynamic sandbox and behavioral analysis approaches that can observe the runtime activity of suspicious files.Integration of Additional Threat Intelligence APIs: In addition to the VirusTotal API, future work could explore integrating other open-source or publicly available threat intelligence services such as Shodan [55] or Censys [56]. These platforms can provide enhanced visibility into potential vulnerabilities, attacker behavior, and exposed services across the monitored network, thereby enriching the honeypot’s analytical depth.Incorporation of Machine Learning and AI: Leveraging machine learning (ML) and artificial intelligence (AI) techniques could enable HoneyLite to adapt dynamically to complex and evolving threat patterns. In particular, anomaly detection methods may help uncover zero-day attacks or novel malware that bypass traditional signature-based detection. Future work will explore unsupervised clustering of attack traffic and predictive modeling to proactively recognize emerging threats.Multi-platform and Container-based Deployment: To increase portability and scalability, HoneyLite could be containerized using Docker or deployed on orchestrated environments like Kubernetes. This would allow easier deployment across diverse infrastructures, including cloud platforms and edge devices. Recent studies show that Docker provides lightweight, portable deployment for security tools [57,58], while Kubernetes enables orchestrated scaling of distributed services [59]. Furthermore, container-based deployment facilitates the creation of distributed honeypot nodes, which can better handle high-concurrency and high-bandwidth attack scenarios. This direction not only addresses scalability challenges but also enables systematic experimentation with performance bottlenecks in adversarial conditions, a limitation noted in our current work.Automated Incident Response Mechanisms: Incorporating automated response features such as dynamic firewall rule updates, isolation of compromised systems, or alerting via email can help in mitigating threats more quickly. Coupling detection with response closes the loop and enhances real-time defense posture.

By pursuing these directions, the HoneyLite honeypot can evolve into a more intelligent, adaptive, and comprehensive security tool, further empowering Small and Medium-sized Enterprises to proactively defend against sophisticated cyber threats.

## 8. Conclusions

This paper introduced HoneyLite, a lightweight and dynamic honeypot-based security system designed to address the specific cybersecurity challenges faced by Small and Medium-sized Enterprises. By integrating real-time network traffic analysis with automated malware detection through the VirusTotal API, HoneyLite effectively detects a wide spectrum of cyber threats, including TCP stealth scans, FTP/SSH intrusions, ICMP flood attacks, and malicious file uploads. Validated within a controlled and simulated virtualized environment, the system demonstrated its ability to operate with minimal resource overhead while producing detailed and structured threat reports for effective incident response and post-event analysis. The results confirm that HoneyLite provides a practical, resource-efficient, and scalable solution for SMEs seeking to enhance their cyber defense posture. Through its affordability, adaptability, ease of deployment, and comprehensive threat visibility, HoneyLite contributes to closing the cybersecurity gap for under-resourced organizations, enabling timely detection, analysis, and mitigation of evolving and sophisticated cyber threats.

## Figures and Tables

**Figure 1 sensors-25-05207-f001:**
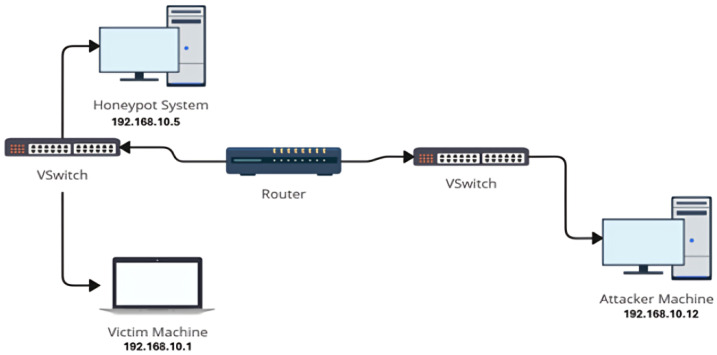
Virtual environment for HoneyLite deployment, simulating attacker, victim, and honeypot nodes. The honeypot intercepts and analyzes malicious traffic to detect and log intrusion attempts in real time. For clarity, the figure is simplified and does not explicitly show subnet masks or network addresses. In practice, proper IP addressing and subnet configurations were applied, ensuring that the router meaningfully separates broadcast domains between attacker, victim, and honeypot nodes.

**Figure 2 sensors-25-05207-f002:**
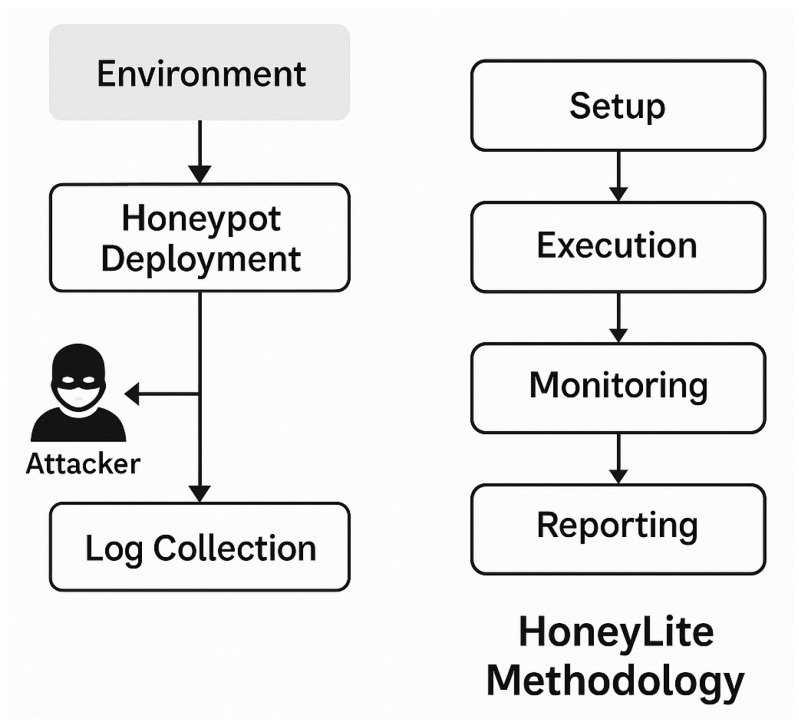
The proposed HoneyLite methodology consists of four abstracted phases: (1) Setup, where the honeypot environment and configurations are prepared; (2) Execution, in which the system is deployed to interact with potential attackers; (3) Monitoring, where malicious activities and attacker behaviors are continuously observed; (4) Reporting, which consolidates findings into actionable insights. These phases provide a streamlined yet flexible workflow for lightweight honeypot deployment and analysis.

**Figure 3 sensors-25-05207-f003:**
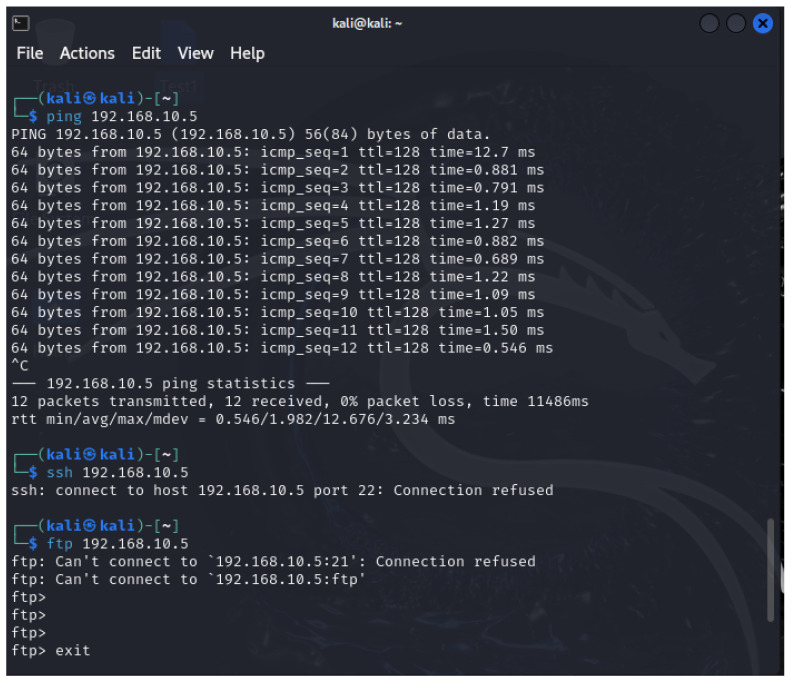
Simulated attack attempts from the Kali Linux machine using ICMP ping, SSH, and FTP connections to probe the HoneyLite honeypot system.

**Figure 4 sensors-25-05207-f004:**
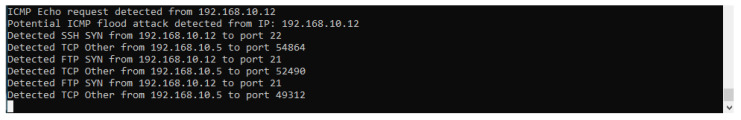
Detection of connection attempts to the victim machine over FTP, SSH, and TCP.

**Figure 5 sensors-25-05207-f005:**
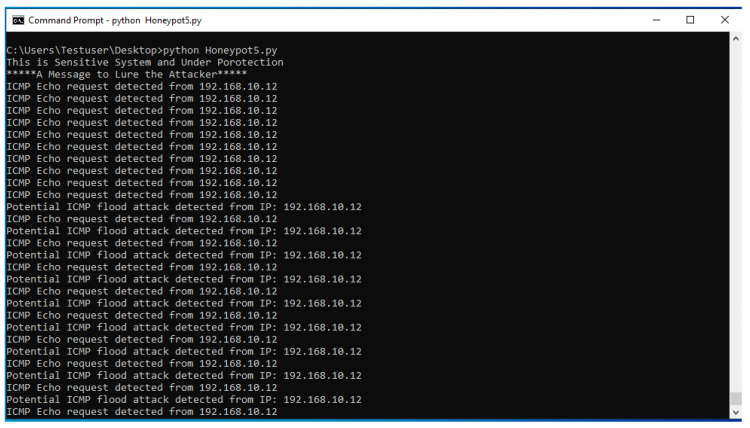
Detection of Denial-of-Service (DoS) attacks.

**Figure 6 sensors-25-05207-f006:**
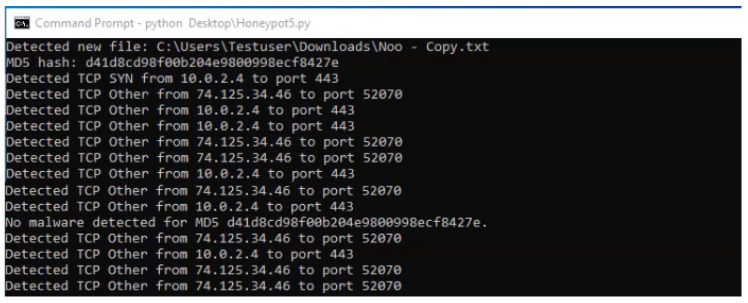
Testing of malware file detection in the system.

**Figure 7 sensors-25-05207-f007:**
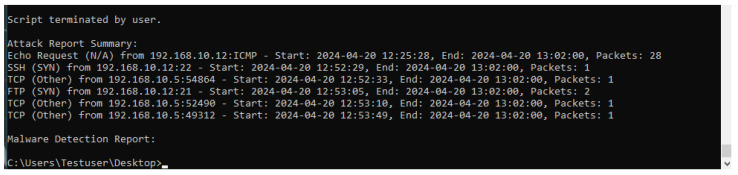
Report generated by the honeypot system.

**Table 1 sensors-25-05207-t001:** Top Risks for Small Enterprises (2018–2023).

Characteristic	2018	2019	2020	2021	2022	2023
Cyber incidents (e.g., cyber crime, malware/ransomware, data breaches)	30%	32%	35%	35%	39%	31%
Macroeconomic developments (e.g., inflation, deflation, monetary policies)	–	–	14%	15%	15%	28%
Energy crisis (e.g., supply shortage/outage, price fluctuations)	–	–	–	–	–	23%
Business interruption (incl. supply chain disruption)	33%	26%	28%	34%	32%	23%
Changes in legislation and regulation (e.g., trade wars, sanctions)	22%	30%	29%	21%	21%	20%

## Data Availability

The original contributions presented in this study are included in the article. Further inquiries can be directed to the corresponding author.

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
