# Peer review of "HoneyLite: A Lightweight Honeypot Security Solution for SMEs"

_sensors, 2025, doi:10.3390/s25165207_

Round 1

Reviewer 1 Report

Comments and Suggestions for Authors

The topic is original and relevant to the answer to security problems in Small and Medium-Sized Enterprises as it offers improvements to the already known Honeypot approach. It is known that there are many security approaches with Honeypot, but these approaches are aged over time and need to be improved to ensure computer network security. 

There is many places in the text of paper where the spaces between signs are not places - page 1 rows 25, 27, 29, 32, 34; page 2 rows 38, 47, 49; page 4 rows 91, 94, 97, 100, 106, 109, 110, 114, 118, 123, 126, 128, 130; page 5 rows 134, 136, 139, 144, 146, 147.

For Figure 1 on page 2 - It is good to present the masks of the used IP addresses and the addresses of the networks in Figure 1, because from the presented visualization it can be concluded that all the end machines (attacker, victim and honeypot) work in one network, although there is a router that divides the network into broadcast domains. In this configuration, the presence of a router becomes meaningless.

For Table 1 on page 3 - Table 1 is not formatted according to the template, which indicates that tables should only have horizontal lines. The text "Cyber incidents (e.g., cyber crime, malware/ransomware, data breaches)" in Table 1 extends beyond the cell it is placed in. It needs to be corrected.

For all abbreviations on the paper - The description of the abbreviation SME is repeated many times, which is unnecessary. The description of the abbreviations is presented when they are first encountered and then only the abbreviations are used in the text. It is recommended to use this approach. This is valid to all abbreviations.

On page 4 row 120 - The description of the abbreviation IoT is not provided.

On page 4 row 122 - The description of the abbreviation AI is not provided.

Citation of references on the text - The template does not indicate that et al citations can be used. It is recommended to check the template again.

On page 6 - It is good to present a block diagram of work of the proposed HoneyLite system.

For part 5.1 Testing Environment on page 8 - It is not described what environment the virtualization is performed with. It is not clear whether virtualization or simulation is used. The configuration of the three roles of the machines Attacker, Victim, Honeypot is not specified.

References of the figure according to the template should be written as "Figure", but it is not used this on page 8 rows 263, 279; page 10 rows 311, 317, 324, 331; page 11 rows 345, 356.

On page 9 rows 292, 293 - No figure is presented to prove that a log of the described information is being implemented.

On page 12 Figure 6 - The information presented in Figure 6 is old and needs to be updated.

For footnotes on the paper - The template does not specify that footnotes can be used, it is recommended to review the template again.

On page 13 row 453 - It is not cited as required in the template. 

Please look at the attached file, and correct the document according to written comments.

Author Response

We thank the reviewer for their highly valuable feedback. Responses are attached. 

For research article: HoneyLite: A Lightweight Honeypot Security Solution for SMEs

Response to Reviewer One Comments

1. Summary

We thank Reviewer 1 for their thorough review and for recognizing the originality and relevance of our work in addressing security challenges in Small and Medium-Sized Enterprises (SMEs) through improvements to the honeypot approach. We have carefully considered each of the comments provided and have revised the manuscript accordingly. All changes are indicated in the revised version.

2. Questions for General Evaluation

Reviewer’s Evaluation

Response and Revisions

Does the introduction provide sufficient background and include all relevant references?

Yes/Can be improved/Must be improved/Not applicable

We thank the reviewer for this observation. We have carefully reviewed all figures and tables to improve their clarity and presentation. These improvements should enhance the clarity and overall presentation of figures and tables in the revised manuscript.

Are all the cited references relevant to the research?

Yes/Can be improved/Must be improved/Not applicable

Is the research design appropriate?

Yes/Can be improved/Must be improved/Not applicable

Are the methods adequately described?

Yes/Can be improved/Must be improved/Not applicable

Are the results clearly presented?

Yes/Can be improved/Must be improved/Not applicable

Are the conclusions supported by the results?

Yes/Can be improved/Must be improved/Not applicable

Are all figures and tables clear and well-presented?

Yes/Can be improved/Must be improved/Not applicable

3. Point-by-point response to Comments and Suggestions for Authors

Comment 1: The topic is original and relevant to the answer to security problems in Small and Medium-Sized Enterprises as it offers improvements to the already known Honeypot approach. It is known that there are many security approaches with Honeypot, but these approaches are aged over time and need to be improved to ensure computer network security.

Response 1: We thank the reviewer for acknowledging the originality and relevance of our work. We agree that traditional Honeypot approaches have become outdated, and our study addresses this by proposing enhanced Honeypot mechanisms tailored to the security needs of SMEs. Our improvements focus on making the system more adaptive, scalable, and resource-efficient, ensuring stronger resilience against modern threats.

Comment 2: There is many places in the text of paper where the spaces between signs are not places - page 1 rows 25, 27, 29, 32, 34; page 2 rows 38, 47, 49; page 4 rows 91, 94, 97, 100, 106, 109, 110, 114, 118, 123, 126, 128, 130; page 5 rows 134, 136, 139, 144, 146, 147.

Response 2: Thank you for pointing this out. We carefully reviewed the manuscript and corrected all spacing inconsistencies.

Comment 3: For Figure 1 on page 2 - It is good to present the masks of the used IP addresses and the addresses of the networks in Figure 1, because from the presented visualization it can be concluded that all the end machines (attacker, victim and honeypot) work in one network, although there is a router that divides the network into broadcast domains. In this configuration, the presence of a router becomes meaningless.

Response 3:  We thank the reviewer for this insightful observation. We acknowledge that the visualization may suggest that all machines operate within a single broadcast domain. However, our intention with Figure 1 was to provide a simplified high-level view of the setup, focusing on illustrating component relationships rather than presenting detailed IP addressing and subnet masks. To avoid confusion, we will clarify this in the text description accompanying Figure 1, highlighting that subnetting and routing are properly configured in the actual deployment, ensuring the router’s role remains meaningful.

Comment 4: For Table 1 on page 3 - Table 1 is not formatted according to the template, which indicates that tables should only have horizontal lines. The text "Cyber incidents (e.g., cyber crime, malware/ransomware, data breaches)" in Table 1 extends beyond the cell it is placed in. It needs to be corrected.

Response 4: We sincerely thank the reviewer for this helpful comment. The table has been reformatted to follow the template requirements by removing vertical lines and using only horizontal rules between rows. In addition, the text in the first row has been wrapped to ensure it no longer extends beyond the cell boundary. These changes have now been applied in Table 1.

Comment 5: For all abbreviations on the paper - The description of the abbreviation SME is repeated many times, which is unnecessary. The description of the abbreviations is presented when they are first encountered and then only the abbreviations are used in the text. It is recommended to use this approach. This is valid to all abbreviations. 

Response 5: The manuscript has been revised so that each abbreviation is defined only at its first occurrence, and thereafter only the abbreviation is used consistently throughout the text.

Comment 6: On page 4 row 120 - The description of the abbreviation IoT is not provided. On page 4 row 120 - The description of the abbreviation AI is not provided.

Response 6: Thank you for highlighting this oversight. We have updated the text.

Comment 7: Citation of references on the text - The template does not indicate that et al citations can be used. It is recommended to check the template again.

Response 7:  We have revised the references in the text to align with the prescribed format.

Comment 8: On page 6 - It is good to present a block diagram of work of the proposed HoneyLite system.

Response 8:  We have included a block diagram (Page 6 - Figure 2)  illustrating the HoneyLite system’s architecture and workflow. This figure provides a clear visual representation of the system’s components and data flow.

Comment 9: For part 5.1 Testing Environment on page 8 - It is not described what environment the virtualization is performed with. It is not clear whether virtualization or simulation is used. The configuration of the three roles of the machines Attacker, Victim, Honeypot is not specified.

Response 9: We thank the reviewer for this valuable suggestion. We have clarified in Section 5.1 that the experiments were conducted in a virtualized environment (VirtualBox) using Linux-based operating systems. Each VM was assigned a specific role (Attacker, Victim, Honeypot), which allowed proper traffic redirection and isolated evaluation. We described the setup at a general level to maintain clarity and ensure the methodology can be understood and potentially replicated across different system configurations.

Comment 10: References of the figure according to the template should be written as "Figure", but it is not used this on page 8 rows 263, 279; page 10 rows 311, 317, 324, 331; page 11 rows 345, 356.

Response 10: Thank you for pointing this out. We have revised all figure references to comply with the template requirements, ensuring that “Figure” is consistently used

Comment 11: On page 9 rows 292, 293 - No figure is presented to prove that a log of the described information is being implemented.

Response 11: We thank the reviewer for the helpful observation. In the revised manuscript, we have included a reference to a public GitHub repository containing the code of the monitoring engine (Page 9 - Section 5). This code demonstrates how metadata such as packet flags, frequency, source and destination IPs, and targeted ports are logged in practice. We believe that providing the source code offers stronger evidence than a figure, while also enhancing the reproducibility and transparency of our work.

Comment 12: On page 12 Figure 6 - The information presented in Figure 6 is old and needs to be updated.

Response 12: We sincerely thank the reviewer for this valuable comment. While we are unable to update Figure 6 in the current version, we have included the full code (Page 9 - Section 5) and methodology, enabling readers to easily reproduce the figure with the latest data at any time.

Comment 13: For footnotes on the paper - The template does not specify that footnotes can be used, it is recommended to review the template again.

Response 13: We have replaced all footnotes with proper references to ensure consistency and compliance with the template guidelines.

Comment 14: On page 13 row 453 - It is not cited as required in the template.

Response 14: Thank you for pointing out this. We have fixed the citation.

Reviewer 2 Report

Comments and Suggestions for Authors

HoneyLite is specifically designed to address the resource constraints and security needs of small and medium-sized enterprises (SMEs), filling the gap left by traditional honeypot systems in terms of resource efficiency and functional adaptability. Its lightweight architecture and use of open-source tools significantly lower the deployment barrier, aligning well with the technical and budgetary limitations of SMEs, thus demonstrating its targeted design. By employing a dual-layer threat detection mechanism combining real-time network traffic analysis with automated malware detection, HoneyLite achieves coordinated defense at both the network and file layers. This design effectively covers common attack types faced by SMEs. The system generates structured threat reports, providing SMEs with clear response guidelines. Furthermore, the virtualized deployment environment and modular design lay the foundation for future scalability, highlighting its operational feasibility and extensibility. Multiple attack scenarios were simulated in a controlled virtual environment, and the system's effectiveness was validated through visualized results, preliminarily demonstrating a balanced trade-off between detection accuracy and resource consumption.Regarding the methodology: the interaction level of the honeypot is ambiguous—the paper does not clearly specify whether HoneyLite is a low-interaction or high-interaction honeypot. Low-interaction honeypots may fail to detect sophisticated attacks, while high-interaction honeypots could increase resource consumption and exposure risks. Additionally, reliance on the VirusTotal API may introduce latency or dependency issues. It is recommended to supplement the analysis with the feasibility of offline detection mechanisms or local sandboxing.In terms of experimental analysis: the results demonstrate HoneyLite's capability to detect specific attacks, but lack quantitative metrics such as detection rate, false positive rate, and false negative rate. It is recommended to include comparative experimental data with existing honeypot systems. While the conclusion emphasizes HoneyLite’s suitability for SMEs, it does not discuss potential performance bottlenecks when facing high-bandwidth or distributed attacks.Overall, the paper excels in methodological innovation, experimental design, and practical application value, exhibiting strong academic and practical significance. I have carefully reviewed your research work and provide the following comments, which I hope will assist you in improving your manuscript:

  1. Supplement data on detection rate, false positive rate, and resource consumption, and conduct comparative experiments with existing solutions to highlight HoneyLite's advantages. Since the current experiments are based on a virtual environment, it is recommended to include deployment cases in real SME networks to analyze the impact of actual interference on detection effectiveness.
  2. Dependence on VirusTotal's known signature database limits the detection of zero-day exploits or novel malware. Consider introducing a dynamic sandbox analysis or behavioral analysis module to monitor runtime behavior of suspicious files. Explore the integration of machine learning (ML) models to identify unknown threats through anomaly detection patterns.
  3. Investigate container-based deployment solutions (e.g., Docker/Kubernetes) to enhance system portability and scalability. Design distributed honeypot nodes to handle high-concurrency attacks.
  4. While the literature review covers over 20 studies, it lacks references to recent research (post-2023) on emerging threats such as ransomware and supply chain attacks. It is recommended to include the latest studies to reflect the timeliness of the research.
  5. Honeypots may involve privacy and compliance issues (e.g., whether capturing attacker data complies with GDPR). It is advisable to include a discussion on relevant regulations or propose compliance-oriented design considerations.

Author Response

We thank the reviewer for their valuable feedback. Responses are attached. 

For research article: HoneyLite: A Lightweight Honeypot Security Solution for SMEs

Response to Reviewer Two

Comments

1. Summary

We thank Reviewer 2 for recognizing the methodological innovation, experimental design, and practical application value of HoneyLite. We have addressed all comments in detail to improve the clarity, completeness, and impact of the manuscript. All changes are marked in the revised version.

2. Questions for General Evaluation

Reviewer’s Evaluation

Response and Revisions

Does the introduction provide sufficient background and include all relevant references?

Yes/Can be improved/Must be improved/Not applicable

We thank the reviewer for pointing out that the description of methods could be improved. In the revised manuscript, we have expanded the methodology section to provide clearer details on the experimental setup and deployment environment. Specifically, we now describe that HoneyLite can be implemented on a standard personal laptop (e.g., 8 GB RAM, quad-core CPU, 256 GB storage), ensuring reproducibility and practical applicability for SMEs without requiring specialized hardware. This addition enhances transparency and makes the methodology more accessible to readers.

Are all the cited references relevant to the research?

Yes/Can be improved/Must be improved/Not applicable

Is the research design appropriate?

Yes/Can be improved/Must be improved/Not applicable

Are the methods adequately described?

Yes/Can be improved/Must be improved/Not applicable

Are the results clearly presented?

Yes/Can be improved/Must be improved/Not applicable

Are the conclusions supported by the results?

Yes/Can be improved/Must be improved/Not applicable

3. Point-by-point response to Comments and Suggestions for Authors

Comment 1: Supplement data on detection rate, false positive rate, and resource consumption, and conduct comparative experiments with existing solutions to highlight HoneyLite's advantages. Since the current experiments are based on a virtual environment, it is recommended to include deployment cases in real SME networks to analyze the impact of actual interference on detection effectiveness.

Response 1: We sincerely thank the reviewer for this valuable suggestion. We fully agree that supplementing quantitative metrics such as detection rate, false positive rate, and resource consumption, alongside comparative experiments with existing honeypot systems, would provide a stronger empirical basis for highlighting HoneyLite’s advantages. However, our current evaluation is intentionally scoped as a proof-of-concept in a controlled virtualized environment, with simplified scenarios representative of SME network conditions. Due to the lightweight design and limited scope of our experiments, we acknowledge that they do not provide a statistically robust basis for generating comprehensive quantitative metrics or for conducting fair comparisons with more complex honeypot frameworks. That said, HoneyLite was designed with minimal configuration overhead and low resource consumption in mind. From our experience in the conducted experiments, we believe that if HoneyLite were subjected to higher-volume or distributed attack scenarios, its lightweight structure should still make it adequate for SMEs, although empirical validation of this claim remains part of our future work. In the revised manuscript, we have clarified this scope in the discussion section and explicitly noted that comparative large-scale experiments and real SME network deployments will be pursued in future work to address this important point.

Comment 2: Dependence on VirusTotal's known signature database limits the detection of zero-day exploits or novel malware. Consider introducing a dynamic sandbox analysis or behavioral analysis module to monitor runtime behavior of suspicious files. Explore the integration of machine learning (ML) models to identify unknown threats through anomaly detection patterns.

Response 2:  We sincerely thank the reviewer for this valuable observation. We acknowledge that the current implementation of HoneyLite relies primarily on VirusTotal’s signature-based detections, which naturally limits its ability to identify zero-day exploits and novel malware strains. This design choice was motivated by the objective of maintaining simplicity and accessibility for SMEs with limited resources.  As the reviewer rightly points out, complementary techniques such as sandbox-based runtime analysis and behavioral profiling would significantly enhance HoneyLite’s detection capabilities. In future work, we plan to integrate a lightweight sandbox module that can safely execute suspicious files and extract behavioral indicators. Moreover, we recognize the potential of anomaly-based detection using machine learning to identify patterns beyond known signatures. We envision extending HoneyLite with a modular ML-based component that leverages anomaly detection techniques for improved zero-day threat detection.  At this stage, our focus was on showing that HoneyLite is practical and easy to use in limited-resource environments, rather than building a complete malware analysis system. Nonetheless, we agree with the reviewer that these enhancements represent an important direction for advancing HoneyLite, and we will explicitly highlight this in the Limitations and Future Work section.

Comment 3: Investigate container-based deployment solutions (e.g., Docker/Kubernetes) to enhance system portability and scalability. Design distributed honeypot nodes to handle high-concurrency attacks.

Response 3: We thank the reviewer for this valuable suggestion. To address this point, we revised the Future Work section (Page 14 - Section 8)  to explicitly discuss container-based deployment using Docker and Kubernetes. We highlighted that containerization improves portability and scalability across cloud and edge environments. Furthermore, we extended the section to emphasize the design of distributed honeypot nodes, which can handle high-concurrency and high-bandwidth attack scenarios more effectively. These additions directly acknowledge the current limitation of our work and outline a concrete research direction.

Comment 4: While the literature review covers over 20 studies, it lacks references to recent research (post-2023) on emerging threats such as ransomware and supply chain attacks. It is recommended to include the latest studies to reflect the timeliness of the research.

Response 4: We thank the reviewer for this valuable observation. In response, we have expanded the Literature Review section by adding a new subsection titled “Emerging Threats: Ransomware and Supply Chain Attacks.” This subsection incorporates and critically discusses several recent works (2024–2025) that directly address the evolving landscape of ransomware and supply chain attacks. These updates ensure that the literature review not only reflects traditional and honeypot-based detection strategies but also addresses the most recent and pressing cybersecurity threats.

Comment 5: Honeypots may involve privacy and compliance issues (e.g., whether capturing attacker data complies with GDPR). It is advisable to include a discussion on relevant regulations or propose compliance-oriented design considerations.

Response 5: We thank the reviewer for this insightful comment. In response, we have added a dedicated subsection (Page 13 - Section 5.3) discussing the privacy, legality, and compliance issues associated with honeypot deployment.
